# Naltrexone Use in Treating Hypersexuality Induced by Dopamine Replacement Therapy: Impact of *OPRM1* A/G Polymorphism on Its Effectiveness

**DOI:** 10.3390/ijms21083002

**Published:** 2020-04-24

**Authors:** Audrey Verholleman, Caroline Victorri-Vigneau, Edouard Laforgue, Pascal Derkinderen, Celine Verstuyft, Marie Grall-Bronnec

**Affiliations:** 1Addictology and Psychiatry Department, CHU Nantes, 44093 Nantes, France; audrey.verholleman@chu-nantes.fr (A.V.); edouard.laforgue@chu-nantes.fr (E.L.); 2Inserm UMR-1246, Université de Nantes, Université de Tours, 44200 Nantes, France; caroline.vigneau@chu-nantes.fr; 3Pharmacology Department, CHU Nantes, 44093 Nantes, France; 4Neurology Department, CHU Nantes, 44093 Nantes, France; pascal.derkinderen@chu-nantes.fr; 5Inserm UMR-1235, Université de Nantes, 44035 Nantes, France; 6Inserm UMR-1178, CESP, Université Paris-Sud, 94276 Le Kremlin Bicêtre, France; celine.verstuyft@aphp.fr; 7Assistance Publique-Hôpitaux de Paris, Service de Génétique moléculaire, Pharmacogénétique et Hormonologie, Hôpitaux Universitaires Paris-Sud, Hôpital de Bicêtre, 94275 Le Kremlin Bicêtre, France

**Keywords:** Parkinson’s disease, dopamine replacement therapy, hypersexuality, impulse control disorder, sex addiction, naltrexone, genotyping, *OPRM1*

## Abstract

Hypersexuality is a well-known adverse side effect of dopamine replacement therapy (DRT), and anti-craving drugs could be an effective therapeutic option. Our aim was to update the knowledge on this issue, particularly on the influence of an Opioid Receptor Mu 1 (*OPRM1*) genetic polymorphism. A systematic review was conducted according to the Preferred Reporting Items for Systematic Reviews and Meta-Analyses (PRISMA) statement. We also analyzed a case of iatrogenic hypersexuality that occurred in a patient treated with DRT. An analysis of the *OPRM1* gene was performed on said patient. Our search identified 597 publications, of which only 7 were included in the final data synthesis. All seven publications involved naltrexone use. Five of them were case reports. None of the publications mentioned DRT side effects, nor did they report genetic data. Regarding our case report, the introduction of naltrexone corresponded with the resolution of the patient’s hypersexuality. Moreover, the patient carried the A/G genotype, which has been reported to be associated with a stronger response to naltrexone for patients with an alcohol use disorder. Although studies are inconclusive so far, naltrexone could be an interesting therapeutic option for resistant hypersexuality due to DRT. Carrying the A/G genotype could help explain a good response to treatment.

## 1. Introduction

Impulse control disorders (ICDs) are frequently found in patients treated for Parkinson’s disease (PD), and are now known to be a relatively frequent side effect of dopamine replacement therapy (DRT) [1,2]. ICDs include different impulsive behaviors, such as pathological gambling, hypersexuality, binge eating, or compulsive shopping [3,4,5]. They share symptoms found in the field of addiction, particularly the failure to reduce or control a behavior despite impaired daily functioning and the resulting negative impact on one’s quality of life [1,6]. The lifetime prevalence of hypersexuality is 2.7% for PD patients on DRT and 7.4% in PD patients treated with dopamine agonists (DAAs) [7]. This prevalence is probably underestimated due to the difficulty of addressing this topic for many patients [8].

There is no currently approved pharmacological treatment for ICDs in PD patients. Nevertheless, a few studies have focused on the effectiveness of opioid antagonists—approved for alcohol dependence treatment—in the treatment of ICDs for patients with PD [9,10,11], especially those with gambling disorders [12,13]. Papay et al. [9] published the results of a placebo-controlled study that enrolled 50 patients with PD and ICDs. They concluded that naltrexone was not superior to the placebo when the Clinical Global Impression-Change scale was used as the effectiveness criteria, but it was superior when the Questionnaire for Impulsive–Compulsive Disorders in Parkinson’s Disease Rating Scale was used. Bosco et al. [12] published three case reports of PD patients with pathological gambling who responded to naltrexone, with resolution of symptoms for all three of them. The effectiveness of these drugs in alcohol use disorders seems to be linked to a genetic polymorphism of the Mu opioid (Mu) receptor (MOR), A118G, although this is still debated [14,15,16]. Although no evidence of effectiveness has been provided for hypersexuality symptoms, opioid antagonists are an interesting therapeutic option, one that is supported by neurobiological data [17,18,19,20]. Indeed, the mesocorticolimbic dopamine pathway, which is known to mediate the reward system, has been implicated in addiction neurobiology, and the involvement of DRT in the emergence of ICDs in PD patients reinforces this hypothesis [1,21]. The opioid system can modulate dopaminergic pathways. Current evidence points to overlaps between addictions—overlaps between neurobiological mechanisms, epidemiology, comorbidities, genetic contributions, etc.—and this leads to the concept of pan-addiction treatment schemes [22,23]. Moreover, some studies seem to have identified that having a specific single nucleotide polymorphism (SNP) in some opioid receptors (especially the Opioid Receptor Mu 1: *OPRM1*) could be a risk factor for developing ICDs [2]. However, to the best of our knowledge, only one publication (written in Dutch and therefore not included in our study) has described the effectiveness of naltrexone in a PD patient with DRT-induced hypersexuality [24].

In the present work, we aimed to update knowledge on this issue, studying in particular the influence of *OPRM1* genetic polymorphisms on naltrexone’s effectiveness. Therefore, we decided to conduct a systematic review on the use of opioid antagonists in the treatment of hypersexuality and to report the case of a patient who developed hypersexuality symptoms while receiving DRT for his PD. These symptoms disappeared after naltrexone was introduced.

## 2. Material and Methods

### 2.1. Systematic Review

#### 2.1.1. Search Strategy

A systematic review of the available literature was conducted to identify all relevant publications using PubMed and ScienceDirect from inception to January 2020. For this review, we complied with the Preferred Reporting Items for Systematic Reviews and Meta-Analyses (PRISMA) guidelines [25].

The search terms were a combination of the following keywords and medical subject heading (MeSH) (United States National Library of Medicine, Bethesda, USA) terms found in the title, abstract, or keywords: “nalmefene” OR “naltrexone” OR “naloxone” AND “hypersexuality” OR “sexuality” OR “sex” OR “sex addiction” OR “compulsive sexuality” OR “impulsive sexuality” OR “sexual behavior” OR “craving”. Duplicates were eliminated. Additional records were included after manual search. The search strategy is summarized in Figure 1.

#### 2.1.2. Eligibility Criteria

Articles had to fulfil the following criteria to be included:The targeted problem was hypersexuality;The medication was an opioid antagonist;The article involved human beings; andThe full article was either in English or French.

#### 2.1.3. Article Selection

Firstly, articles were selected based on their titles and abstracts. Secondly, the full text of all the included articles was read. The authors (Audrey Verholleman and Marie Grall-Bronnec) performed this work independently using the same bibliographic search. If the authors disagreed about the relevance of an article, it was discussed.

#### 2.1.4. Data Extraction

Clinical and genetic data were extracted from the articles. The factors considered included study design, sample size, participants and hypersexuality characteristics, drugs taken, and objectives.

### 2.2. Case Report

We also report a case of iatrogenic hypersexuality that occurred in a patient treated with DRT. An OPRM1 gene analysis was performed.

## 3. Results

### 3.1. Systematic Review

Of the potential 597 articles, 7 met the criteria for inclusion. All involved naltrexone use. Five of them were case reports, one was a retrospective study, and one was an open-ended prospective study. Regarding the case reports, six patients with compulsive sexual behavior symptoms were described. Five were male, one was female. They were treated with naltrexone with a positive outcome. Most patients had tried psychotherapy and antidepressants with no significant results. In each case, the introduction of naltrexone was quickly followed by a decrease in symptom intensity, and each patient reported a long-lasting remission. Three patients had had adjuvant therapy using serotonin reuptake inhibitors, without any change during the months preceding naltrexone introduction. Both the retrospective study and the prospective study (including 40 patients in total, all male) resulted in a clinical improvement with naltrexone use for most of the included patients. Naltrexone was not associated with any side effects. No articles mentioned side effects of DRT or reported genetic data.

The results are summarized in Table 1.

### 3.2. Case Presentation

Patient A is a 66 year old Caucasian male, married with two children and unemployed. He was referred to the addiction department by his neurologist in 2016 for hypersexuality symptoms. Except for his PD, his past medical history involved several depressive episodes, for which he was treated with sertraline for several years. He also experienced anxiety disorders (social anxiety disorder and generalized anxiety disorder) in his thirties. No comorbid addictive disorders were reported except for his current active tobacco smoking.

He was diagnosed with PD in 2006 and was first treated with pramipexole, a DAA, in 2007. This therapy was effective and well tolerated for several years with dosages between 1.4 and 2.1 mg per day. Rasagiline, a selective monoamine oxidase (MAO) B inhibitor, was added in 2010 to enhance the dopaminergic correction. This treatment was stopped in 2012, and levodopa (L-dopa)/benserazide therapy was initiated instead, as explained in our treatment chart (Figure 2).

He reported hypersexuality symptoms to his neurologist for the first time in 2014. He was treated at the time with a DAA and L-dopa. His therapist decided, according to the guidelines, to stop pramipexole treatment in November 2014. After this change, he was treated with a combination of L-dopa and a catechol-O-methyltransferase (COMT) inhibitor, to which rasagiline (MAO B inhibitor) was added again a few months later, in March 2015. Since the symptoms persisted despite stopping the DAA treatment and as the marital repercussions were becoming stronger, Patient A was referred to our addictology unit in February 2016. He was treated at the time with a combination of L-dopa, a COMT inhibitor, and a MAO B inhibitor. The first clinical assessment confirmed the diagnosis of hypersexuality disorder, according to the DSM-IV criteria [26], and of sexual addiction, according to the criteria proposed by Goodman [27]. Symptoms of hypersexuality appeared for the first time in January 2014, and included masturbatory behavior, pornographic video watching, and sexual intercourse with prostitutes. He met most of the criteria for addiction, such as a loss of control, a failure to reduce or stop these behaviors despite the consequences, a growing sense of tension prior to the behavior, agitation and irritability when it was impossible to engage in the behavior, and frequent and time-consuming preoccupation with sex, resulting in impaired functioning.

In addition to psychological support, and after checking that there were no contraindications (especially liver and kidney failure), treatment with naltrexone was introduced in September 2016 at a dosage of 50 mg per day. The patient was informed that he could not use opioid drugs such as codeine, tramadol, or morphine during his treatment with naltrexone. The hypersexuality symptoms quickly disappeared, and at the next appointment, Patient A had totally stopped his addictive sexual behavior and did not report any obsessive thoughts about sex. He no longer had any symptoms of ICD. Treatment with naltrexone was well tolerated, and the patient did not report any adverse side effects.

Another therapeutic adaptation was made concerning his PD treatment with the discontinuation of the MAO B inhibitor at the end of 2016, maintaining only a double-therapy using L-dopa and a COMT inhibitor. The MAO B inhibitor (rasagiline) was briefly reintroduced from January 2017 to March 2017 and then stopped again.

Patient A came to several more appointments in the addiction unit without reporting any signs of ICD and, more particularly, without reporting any signs of hypersexuality, for a period of 17 months from October 2016 to February 2018. This stable period motivated an attempt to stop naltrexone treatment in February 2018.

Approximately two weeks after the treatment interruption, his hypersexuality symptoms reappeared, consisting of obsessive thoughts about sex, masturbation, and visiting pornographic websites. There was no change in the PD treatment during this time, which still consisted of L-dopa and a COMT inhibitor. One week after the reoccurrence of these symptoms, Patient A decided to call our unit to inform us of his relapse and ask for advice. He had already started to resume his medication. Naltrexone (50 mg per day) was thus prescribed again, and Patient A returned for an appointment one week later, after 13 days of naltrexone therapy. He had completely stopped visiting pornographic websites and no longer described a desire to do so. He still reported some thoughts about sex, particularly when he was alone and experiencing psychological tension or worries, but he felt no compulsion to act on those thoughts anymore.

Given the clinical improvement following the resumption of naltrexone, the decision was made to maintain treatment with naltrexone for several months. For a year, Patient A did not report many inconvenient hypersexuality symptoms. He described from time to time having thoughts about sex and experiencing a few urges to visit pornographic websites, but he did not satisfy those urges. These thoughts were more frequent during times of stress or change (for example, during times when his wife was away). There was no change in his PD therapy except for a slight increase in the L-dopa dosage.

In February 2019, another attempt was made to stop naltrexone treatment. It was discontinued on February 25th. Patient A reported the occurrence of obsessive thoughts about sex and a strong craving on March 18th. These hypersexuality symptoms got stronger from that day up to March 25th, when Patient A managed to talk to his wife about them without having acted on them. The following day, the treatment with naltrexone was resumed, and Patient A described a quick improvement with the disappearance of his obsessive thoughts and craving after two days. He is currently still receiving treatment with naltrexone (50 mg per day) and has not reported any cravings or other signs of hypersexuality.

Considering the effectiveness of naltrexone on the hypersexuality symptoms in Patient A, we asked for a genetic analysis of the *OPRM1* gene and found that he was carrying the G allele of the A118G polymorphism and was thus heterozygous for this gene.

## 4. Discussion

Our systematic review showed that only a few articles were available about the potential use of opioid antagonists in treating hypersexuality symptoms, even in the general population, and most of these were case reports. One potential limitation could be the definition of hypersexuality or compulsive sexual behavior; most of the articles did not use validated diagnostic criteria, thus weakening potential comparisons between cases.

Our case report is consistent with the available literature data and useful to strengthen the hypothesis of naltrexone effectiveness. In this case, we used validated criteria to diagnose hypersexuality using criteria from both the DSM and Goodman [26,27]. The occurrence of an ICD such as hypersexuality in a patient treated for PD is probably linked to the DRT, and more particularly to the DAA. Here, the probability that the hypersexuality symptoms were due to pramipexole was rated as “possible” using the Naranjo Scale for Adverse Drug Reaction Probability Scale [28]. Indeed, there are several previous reports of this adverse drug reaction in the literature, and a statistically significant association was found between DAAs and ICDs, including hypersexuality [1]. An even stronger association was found with the use of pramipexole and ropinirole, both DAAs with a preferential affinity for the dopamine D3 receptor [29]. It is surprising that hypersexuality symptoms only appeared after 8 years of treatment with pramipexole without major changes in the dosage. It might have been expected that these symptoms would disappear after the interruption of the treatment with pramipexole, but this was not the case, even if some of the symptoms seemed to decrease in intensity. Moreover, it is important to bear in mind that the patient kept being treated with DRT (L-dopa and rasagiline). Both drugs have been linked, although less frequently than pramipexole, to the development of ICDs [10,30].

This case is especially interesting for its therapeutic implications. To date, no pharmacological agents have been approved with a specific indication in treating ICDs or hypersexuality [31]. In the case of DRT-induced hypersexuality symptoms, the first step is to reduce the dose of dopaminergic medication and, if possible, to switch from DAAs to L-dopa [3,13,24,32]. Most of the time, patients respond to dopaminergic therapy reduction, but sometimes the DRT cannot be modified, or the symptoms persist despite the adjustment [1]. In Patient A’s case, hypersexuality symptoms persisted after the discontinuation of pramipexole. These symptoms finally disappeared shortly after the beginning of the treatment with naltrexone and remained stable over several months. What is even more striking is the reappearance of the same hypersexuality signs during both attempts to stop naltrexone treatment, followed consistently by remission after treatment resumption.

The concept of treating ICDs with opioid antagonists is based on the neurobiological similarities between addictive disorders in general and substance use disorders [23]. The role of the mesocorticolimbic dopaminergic pathway in the reward system has been widely described and strongly implicated in addictions [33]. Drugs or certain behaviors can trigger dopamine release, thus producing a feeling of well-being, and dopamine can reinforce this system by enhancing the rewarding nature of these behaviors, which can lead to the development of addictions [20,22]. The major role of dopamine in the addictive process is reinforced by the possible emergence of addictive behaviors and ICDs in patients receiving drugs acting on the dopamine system, especially dopamine agonists that act directly on dopamine receptors [6].

The opioid system can modulate dopaminergic pathways; we know of three opioid receptors—the mu opioid receptors (MORs), the delta opioid receptors (DORs), and the kappa opioid receptors (KORs)—that are activated by endogenous opioid peptides and by recreational drugs [33,34]. Opioids increase dopamine levels in the brain by inhibiting GABA release (which normally inhibits dopamine release) [35]. They play a major part in the addictive process, especially through the reward system and the reinforcement process. It seems that MORs and, to a lesser extent, DORs are linked to positive reinforcement, while KORs are linked to negative reinforcement [36]. MOR and DOR antagonists are thus supposed to decrease the positive reinforcement of opiate and non-opiate drugs and of natural rewarding behaviors, thus reducing cravings and drug-seeking behaviors. KOR antagonists could have an antidepressant effects in individuals with stress-induced depressive behaviors, ease emotional withdrawal symptoms, and decrease stress-induced drug seeking relapse in animals [36,37].

Naltrexone, a mu delta kappa antagonist, has been approved for the treatment of alcohol dependence [38,39,40], and is increasingly being evaluated as a potential treatment for other addictive disorders, including behavioral addictions and ICDs [9,12]. A few cases of patients presenting hypersexuality symptoms treated with naltrexone have been reported, some with interesting results [41,42]. Our case report strengthens the hypothesis that naltrexone is an interesting substance not only for the management of alcohol use disorder, but also for the management of other addictive disorders, including behavioral addictions.

Using an opioid antagonist for the treatment of ICDs appearing during PD therapy might seem counterintuitive because of its impact on dopamine levels. Indeed, PD is caused by a dopamine deficit in the substantia nigra and the striatum [43]. The purpose of PD treatments such as DAAs or L-dopa is to increase dopamine levels in the nigrostriatal pathway, but they also act in the mesocorticolimbic pathway, causing iatrogenic ICDs [1]. In contrast, the effect of opioid antagonists is to ultimately decrease dopamine levels, but only in the mesocorticolimbic pathway [19,44]. These two opposite effects seem to contraindicate the use of an opioid antagonist during treatment for PD, but no study has mentioned an aggravation of PD symptoms following treatment with naltrexone. In our case report, the introduction of naltrexone did not seem to have any impact on the patient’s PD evolution. Furthermore, the medication was well tolerated. Safety considerations for using naltrexone in PD patients with ICDs include potential physical comorbidities that may interfere with pharmacological effects, distribution and metabolism, and concomitant medication for the treatment of comorbid physical and psychiatric conditions. In Patient A’s case, there were no drug interactions.

Another interesting point in our case report is the result of the pharmacogenetic analysis. As we said, several studies have pointed to the effectiveness of naltrexone for alcohol use disorder, but most of them found interindividual variations in this treatment’s effectiveness. Interindividual variations can be influenced by several factors, including genetic background [45]. Naltrexone binds preferentially to the MOR, which is encoded by the *OPMR1* gene. This gene has thus been particularly studied to determine whether the *OPMR1* polymorphism could have an impact on the patient’s response to naltrexone [46]. *OPRM1* has a functionally significant and common variant termed A118G (*rs1799971*) [47]. This single-nucleotide polymorphism (SNP) in Exon 1 causes the transition of an adenine (A) nucleotide to guanine (G) at Base 118. In turn, A118G causes an amino acid exchange at Residue 40 of the MOR from the normal asparagine (Asn, A allele) to an abnormal aspartic acid (Asp) residue (G allele) (Asn40Asp) [47]. The Asp 40 isoform (G allele) of the receptor reduces the isoform’s expression at cell surface, decreasing the MOR’s binding potential in the brain, and increases the morphine requirement for its activation [47]. A significant reduction in the efficacy of signaling pathways after the binding of specific agonists has been observed [47]. The rate of G coupling in carriers of the G allele is only half that of wild-type AA homozygotes. The frequency of the mutated G allele is 20%, with a wide population-specific range from 3% in Africans to nearly 50% in Asians.

Studies are inconclusive about the potential association of the *OPRM1* A/G polymorphism with addictive disorders. Some studies have shown a protective effect, others have shown an increased risk of developing an addictive disorder, and others found no association. Several meta-analyses, one focusing on general substance dependence [48], two focusing on opiate dependence [49,50], and one focusing on alcohol dependence [51], found no association, whereas the most recent meta-analysis found a moderate protective effect of the *OPRM1* A/G polymorphism against substance abuse in studies conducted in European populations [52]. A recent case–control study among a PD patient cohort suggested a protective effect of the *OPRM1* A118G variant against the risk of developing an ICD while receiving dopamine therapies [53].

It is also suspected that this polymorphism may influence naltrexone’s effectiveness. Despite some inconsistent results, a few studies have reported that among patients with alcohol use disorders treated with naltrexone, those carrying the *OPRM1* A118G variant had a lower relapse rate than those carrying the *OPRM1* A118A variant [54,55]. A meta-analysis conducted in 2012 found that naltrexone-treated patients carrying the G allele had significantly lower relapse rates than those who were homozygous A/A, but found no difference in abstinence rates between the two groups [16]. The difference in relapse rates seemed specific to naltrexone, because no differences in relapse rates were found between *OPRM1* A118G groups and *OPRM1* A118A groups treated with a placebo. It is possible that this increased response to naltrexone could be found in patients suffering from other addictive disorders, substance abuse disorders, or behavior addictions. Knowing which patients are more likely to respond to naltrexone could be useful when choosing therapeutic strategies.

Regarding Patient A, the fact that he carries the *OPRM1* A118G polymorphism could strengthen the hypothesis that his hypersexuality symptoms responded to treatment with naltrexone. More generally, there are not yet enough to conclude that naltrexone is effective when treating hypersexuality symptoms, but the results are promising. Randomized controlled trials are required to further investigate this therapeutic option both in the general population and in specific populations, such as patients treated with DRT. More genetic research should be conducted, especially pharmacogenetic investigations, which could help to predict individual therapeutic responses and help clinicians choose the best treatment for each patient in individualized care.

## Figures and Tables

**Figure 1 ijms-21-03002-f001:**
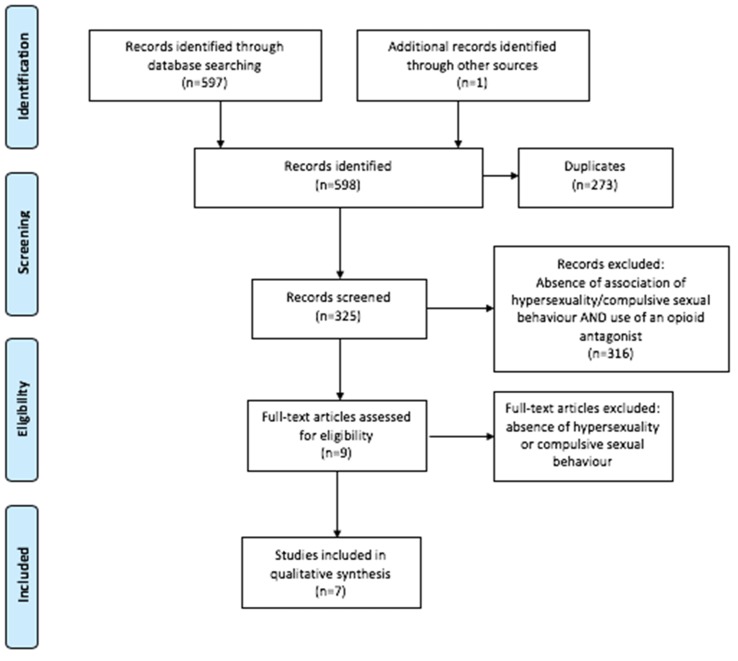
PRISMA 2009 flow diagram: identification, screening, eligibility, and inclusion.

**Figure 2 ijms-21-03002-f002:**
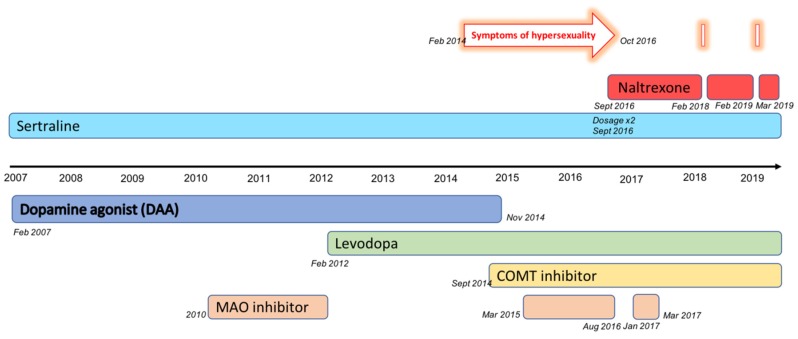
Treatment chart.

**Table 1 ijms-21-03002-t001:** Results of the systematic review.

Title, Authors, Date	Study Design	Sample Size	Characteristics of Participants	Objectives	Methods	Results
A case of kleptomania and compulsive sexual behavior treated with naltrexone Grant, J.E., Kim, S.W. 2001	Case report	*n* = 1	Male patient, 58 years old. Kleptomania since he was 11, and compulsive sexual behavior since his fifties. Antidepressants and psychotherapy for 10 years were ineffective.		(-)	Disappearance with naltrexone of his urges to steal and to have sex (from 25 mg to 150 mg per day). Relapse at the same intensity as before 3 days after the discontinuation of treatment, with a remission 4 days after restarting naltrexone. Stability for the 20 weeks of follow-up with 150 mg of naltrexone per day. Naltrexone was well tolerated. No data about genetics.
Treatment of compulsive sexual behavior with naltrexone and serotonin reuptake inhibitors: two case studies Raymond, N.C., Grant, J.E., Kim, S.W. 2002	Case report	*n* = 2	**Case 1:** A 42 year old woman reporting compulsive sexual behavior, associated with depression and anxiety symptoms. She had an history of cocaine use disorder. Fluoxetine (60 mg/day) was effective on depression and anxiety symptoms but not on sexual urges. **Case 2:** A 62 year old male reporting intermittent compulsive sexual behavior for 20 years. Several antidepressants were ineffective on sexual urges (fluoxetine, bupropion, citalopram, buspirone).		(-)	**Case 1:** Decrease of compulsive sexual behavior 2 weeks after naltrexone initiation (50 mg/day) and almost complete remission of sexual urges at 100 mg/day. Naltrexone was well tolerated. No data about genetics. **Case 2:** Diminution of intrusive thoughts about sex and control over compulsive sexual behavior after one month with naltrexone (50 mg/day). Remission for the 8 months of follow-up with 100 mg of naltrexone per day. Naltrexone was well tolerated. No data about genetics.
Naltrexone in the treatment of adolescent sexual offenders Ryback, R.S. 2004	Open-ended prospective study	*n* = 21	Male adolescents participating in an inpatient adolescent sexual offenders’ program. Inclusion criteria: Masturbating ≥3 times per day; feeling unable to control arousal; spending more than 30% of awake time in sexual fantasies; or interfering in their functioning	To investigate whether naltrexone can decrease sexual arousal	Naltrexone was given for 2 months to all participants, then stopped for 13 of them (according to the initial study design). Monitoring was made using a fantasy-tracking log and a masturbation log. Outcome: over 30% decrease in any self-reported criterion for at least 4 months	Significant clinical improvement for 15 out of 21 patients, with an average dose of 160 mg/day. Dosages above 200 mg/day were not more useful. Discontinuation of naltrexone in 13 of the patients resulted in the reoccurrence of symptoms. No data about genetics.
Internet sex addiction treated with Naltrexone Bostwick, J.M., Bucci, J.A. 2008	Case report	*n* = 1	Male patient who first met a psychiatrist for sexual addiction at age 24 and was followed for 7 years. Diagnosis of sexual addiction defined as compulsive sexual behavior persisting despite serious negative consequences. Antidepressant, individual, and group psychotherapy were ineffective.		(-)	Nearly complete remission for more than three years (time of follow-up) with naltrexone from 50 mg to 150 mg per day. Naltrexone was well tolerated. No data about genetics.
Augmentation with naltrexone to treat compulsive sexual behavior: a case series Raymond, N.C., Grant, J.E., Coleman, E. 2010	Retrospective study	*n* = 19	Male outpatients with compulsive sexual behavior consulting in a sexual health clinic in Minnesota.	To investigate whether naltrexone can reduce urges and compulsive sexual behavior	Treatment with naltrexone. Assessment with a Clinical Global Impression (CGI) scale.	Reduction in compulsive sexual behavior for 17 out of 19 patients (CGI score of 1 or 2, “very much improved” or “much improved”). Mean effective dose for the 17 patients was 104 (+/− 41) mg per day. Naltrexone was well tolerated. No data about genetics.
Treatment of compulsive pornography use with naltrexone: a case report Kraus, S.W., Meshberg-Cohen, S. 2015	Case report	*n* = 1	Male in his thirties with compulsive masturbation to pornography with numerous failed attempts to quit. Effectiveness of several weeks of cognitive-behavioral therapy (CBT) on his use of pornography (−70%) but not on sexual urges.		(-)	Initiation of naltrexone 50 mg/day after 10 weeks of CBT was more effective on craving, with an associated decrease in pornography use. No data about genetics.
Compulsive sexual behaviors treated with naltrexone monotherapy Camacho, M., Moura, A.M., Oliveira-Maia, A.J. 2018	Case report	*n* = 1	27 year old man with compulsive sexual behaviors (significant amount of time and money spent for his fantasies, loss of control, associated with anxiety and depression symptoms). Antidepressants, mood stabilizers, and neuroleptics were ineffective.		(-)	Treatment with fluoxetine and aripiprazole at the time of inclusion. Reduction of sexual fantasies and control increase with naltrexone 50 mg/day and nearly complete remission for 10 months with naltrexone 100 mg/day. Naltrexone was well tolerated. No data about genetics.

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
