# Peer review of "Naltrexone Use in Treating Hypersexuality Induced by Dopamine Replacement Therapy: Impact of OPRM1 A/G Polymorphism on Its Effectiveness"

_ijms, 2020, doi:10.3390/ijms21083002_

Round 1

Reviewer 1 Report

I am extremely impressed with goal and techniques that applied in this manuscript.

In my opinion this manuscript requires only minor grammar and typo errors check and ready for the publication.

Author Response

Answer: We thank the reviewer for his/her kind comments and his compliments.

We apologise for the grammar and typo errors. We received the assistance of Springer Nature Author Services, who had edited the first version of manuscript (please refer editing certificate at the end of the previous cover letter), and we asked for a native English-speaking colleague to check the revised version of the manuscript.

Reviewer 2 Report

The authors partly updated the knowledge about the possible effect of anti-craving drugs as an effective therapeutic options for treatment of hypersexuality, a well-known adverse side effect of dopamine replacement therapy (DRT), particularly the influence of Opioid Receptor Mu 1 (OPRM1) genetic polymorphism. They proposed a systematic review conducted according to the PRISMA statement as well as also analysed the case of iatrogenic hypersexuality that occurred in a patient treated with DRT.

In principle, this topic is interesting and this manuscript could provide an interesting new finding about the possible treatment of hypersexuality as a very unpleasant side symptoms of dopamine replacement therapy in patients with PD. The ms seems to be a  mixture of a review and case report about a single patient.

The authors claim that their findung is a new one. Unfortunately, this is not the case.

In the following the results of our search in the literature. That should have been done by the authors of the ms.

Literature with respect to “Naltrexone and hypersexuality in Parkinson disease”

[Hypersexuality and other impulse control disorders in Parkinson's disease].

[Article in Dutch]

Nelis EA1, Berendse HW, van den Heuvel OA.

Author information

Abstract

Impulse control disorders (ICD) in Parkinson's disease (PD) pose a therapeutic challenge. This article provides a description of the symptoms and management strategies of ICD in PD. We present two men aged 52 and 69 with ICD, especially hypersexuality, in response to dopaminergic medication. In the first case the symptoms of hypersexuality and gambling decreased after reducing the dose of the dopamine-agonist. In the second case the hypersexuality symptoms decreased after addition of naltrexon. It is important to recognize the symptoms of ICD in PD because of the large impact on social and relational functioning. It is of great importance to repeatedly ask the patient and their partner about these symptoms, since feelings of shame and guild hamper spontaneous report. The first step of treatment consists of reducing the dose of dopaminergic medication and/or to switch from dopamine-agonist to levodopa. Although the research on effective treatment options has been limited so far, treatment alternatives from the addiction field seem promising.

PMID:26732226

Neurology. 2014 Aug 26;83(9):826-33. doi: 10.1212/WNL.0000000000000729. Epub 2014 Jul 18.

Naltrexone for impulse control disorders in Parkinson disease: a placebo-controlled study.

Papay K1, Xie SX1, Stern M1, Hurtig H1, Siderowf A1, Duda JE1, Minger J1, Weintraub D2.

OBJECTIVE:

Impulse control disorders (ICDs) in Parkinson disease (PD) are common and can be difficult to manage. The objective of this study was to determine the efficacy and tolerability of naltrexone, an opioid antagonist, for the treatment of ICDs in PD.

METHODS:

Patients with PD (n = 50) and an ICD were enrolled in an 8-week, randomized (1:1), double-blind, placebo-controlled study of naltrexone 50-100 mg/d (flexible dosing). The primary outcome measure was response based on the Clinical Global Impression-Change score, and the secondary outcome measure was change in symptom severity using the Questionnaire for Impulsive-Compulsive Disorders in Parkinson's Disease-Rating Scale (QUIP-RS) ICD score.

RESULTS:

Forty-five patients (90%) completed the study. The Clinical Global Impression-Change response rate difference favoring naltrexone in completers was 19.8% (95% confidence interval [CI] -8.7% to 44.2%). While this difference was not significant (odds ratio=1.6, 95% CI 0.5-5.2, Wald χ2 [df]=0.5 [1], p=0.5), naltrexone treatment led to a significantly greater decrease in QUIP-RS ICD score over time compared with placebo (regression coefficient for interaction term in linear mixed-effects model=-7.37, F[df]=4.3 [1, 49], p=0.04). The estimated changes in QUIP-RS ICD scores from baseline to week 8 were 14.9 points (95% CI 9.9-19.9) for naltrexone and 7.5 points (95% CI 2.5-12.6) for placebo.

CONCLUSIONS:

Naltrexone treatment was not efficacious for the treatment of ICDs in PD using a global assessment of response, but findings using a PD-specific ICD rating scale support further evaluation of opioid antagonists for the treatment of ICD symptoms in PD.

CLASSIFICATION OF EVIDENCE:

This study provides Class I evidence that in patients with PD and an ICD, naltrexone does not significantly increase the probability of achieving response. However, the study lacked the precision to exclude an important difference in response rates.

© 2014 American Academy of Neurology.

Front Neurol. 2019 Apr 17;10:351. doi: 10.3389/fneur.2019.00351. eCollection 2019.

Impulse Control Disorders in Parkinson's Disease. A Brief and Comprehensive Review.

Gatto EM1,2, Aldinio V1.

Author information

Abstract

Impulse control and related disorders (ICDs-RD) encompasses a heterogeneous group of disorders that involve pleasurable behaviors performed repetitively, excessively, and compulsively. The key common symptom in all these disorders is the failure to resist an impulse or temptation to control an act or specific behavior, which is ultimately harmful to oneself or others and interferes in major areas of life. The major symptoms of ICDs include pathological gambling (PG), hypersexualtiy (HS), compulsive buying/shopping (CB) and binge eating (BE) functioning. ICDs and ICDs-RD have been included in the behavioral spectrum of non-motor symptoms in Parkinson's disease (PD) leading, in some cases, to serious financial, legal and psychosocial devastating consequences. Herein we present the prevalence of ICDs, the risk factors, its pathophysiological mechanisms, the link with agonist dopaminergic therapies and therapeutic managements.

KEYWORDS:

ICD; Parkinson disease; binge eating; compulsive buying; hypersexual disorder; impulse control disorders; pathological gambling

PMID:31057473

 DOI:10.3389/fneur.2019.00351

Taking into consideration that specific SNP opioid receptors have been identified as stronger risk factors for ICDs, opioid antagonists employed in the treatment of PG have produced controversial results (naltrexone, nalmefene) (2, 7, 16, 22, 60, 123).

Parkinsonism Relat Disord. 2019 Feb;59:65-73. doi: 10.1016/j.parkreldis.2019.02.042. Epub 2019 Feb 28.

Current treatment of behavioral and cognitive symptoms of Parkinson's disease.

Rektorova I1.

Author information

Abstract

Cognitive and behavioral symptoms are common in Parkinson's disease, may occur even in the prodromal stages of the disease, worsen with disease progression, and surpass motor symptoms as the major factors affecting patient quality of life and caregiver burden. The symptoms may be caused by the disease pathology or they may represent adverse effects of treatment, or both etiological factors may contribute. Although many of these symptoms are related to dopaminergic dysfunction or dopaminergic medication, other neurotransmitters are involved as well. Behavioral symptoms including impulse control disorders, apathy, psychosis, as well as mild cognitive impairment and dementia are reviewed with a special focus on current treatment approaches.

Copyright © 2019. Published by Elsevier Ltd.

KEYWORDS:

Apathy; Behavioral; Cognitive; Dementia; Dopaminergic; Impulse control disorders; Mild cognitive impairment; Parkinson's disease; Psychosis; treatment

PMID:30852149

 DOI:10.1016/j.parkreldis.2019.02.042

Case reports and small case series studies have described possible

effects of neuroleptics, antidepressants, S5αR inhibitor finasteride and

various anticonvulsant drugs; however, there is no clear evidence for

the use of these drugs in the treatment of ICDs; for review, see Refs.

[8,9,15,35,36]. An 8-week, randomized, double-blind, controlled study

investigated the opioid antagonist naltrexone in 50 PD patients with

ICDs [37], and a 17-week study examined the NMDA antagonist

amantadine in 17 PD patients with pathological gambling [38

Naltrexone and gambling in Parkinson disease:

Clin Neuropharmacol. 2012 May-Jun;35(3):118-20. doi: 10.1097/WNF.0b013e31824d529b.

Opioid antagonist naltrexone for the treatment of pathological gambling in Parkinson disease.

Bosco D1, Plastino M, Colica C, Bosco F, Arianna S, Vecchio A, Galati F, Cristiano D, Consoli A, Consoli D.

Author information

Abstract

Pathological gambling (PG) is a potential complication related to the treatment of Parkinson disease (PD) with dopamine agonists (DA). The cause of this disorder is unknown, but altered dopamine neurotransmission may be involved.

OBJECTIVE:

We evaluated the efficacy and tolerability of the opioid antagonist naltrexone in the treatment of PG in PD.

METHODS:

Our cases included 3 patients with PD who developed PG after DA treatment.

RESULTS:

Pathological gambling did not improve after reduction or discontinuation of DA. These patients responded poorly to serotonin reuptake inhibitors, whereas treatment with opioid antagonist naltrexone resulted in the remission of PG. Naltrexone treatment was well tolerated. In one patient, higher dose of naltrexone resulted in hepatic abnormalities, which resolved after dosage reduction.

CONCLUSIONS:

The opioid antagonist naltrexone could be an effective option for the treatment of PG in PD.

PMID:22426027

DOI:10.1097/WNF.0b013e31824d529b

Lancet Neurol. 2017 Mar;16(3):238-250. doi: 10.1016/S1474-4422(17)30004-2. Epub 2017 Feb 15.

Impulse control disorders and levodopa-induced dyskinesias in Parkinson's disease: an update.

Voon V1, Napier TC2, Frank MJ3, Sgambato-Faure V4, Grace AA5, Rodriguez-Oroz M6, Obeso J7, Bezard E8, Fernagut PO8.

Author information

Abstract

Dopaminergic medications used in the treatment of patients with Parkinson's disease are associated with motor and non-motor behavioural side-effects, such as dyskinesias and impulse control disorders also known as behavioural addictions. Levodopa-induced dyskinesias occur in up to 80% of patients with Parkinson's after a few years of chronic treatment. Impulse control disorders, including gambling disorder, binge eating disorder, compulsive sexual behaviour, and compulsive shopping occur in about 17% of patients with Parkinson's disease on dopamine agonists. These behaviours reflect the interactions of the dopaminergic medications with the individual's susceptibility, and the underlying neurobiology of Parkinson's disease. Parkinsonian rodent models show enhanced reinforcing effects of chronic dopaminergic medication, and a potential role for individual susceptibility. In patients with Parkinson's disease and impulse control disorders, impairments are observed across subtypes of decisional impulsivity, possibly reflecting uncertainty and the relative balance of rewards and losses. Impairments appear to be more specific to decisional than motor impulsivity, which might reflect differences in ventral and dorsal striatal engagement. Emerging evidence suggests impulse control disorder subtypes have dissociable correlates, which indicate that individual susceptibility predisposes towards the expression of different behavioural subtypes and neurobiological substrates. Therapeutic interventions to treat patients with Parkinson's disease and impulse control disorders have shown efficacy in randomised controlled trials. Large-scale studies are warranted to identify individual risk factors and novel therapeutic targets for these diseases. Mechanisms underlying impulse control disorders and dyskinesias could provide crucial insights into other behavioural symptoms in Parkinson's disease and addictions in the general population.

Copyright © 2017 Elsevier Ltd. All rights reserved.

PMID:28229895

DOI:10.1016/S1474-4422(17)30004-2

Naltrexone, an opioid antagonist,

decreased symptoms in patients with Parkinson’s

disease, relative to placebo, but did not improve global

symptom severity.101

J Neurol (2015) 262:7–20 DOI 10.1007/s00415-014-7361-4

Impulse control disorders in Parkinson’s disease: an overview from neurobiology to treatment

 Emke Mare´chal • Benjamin Denoiseux • Ellen Thys • David Crosiers • Barbara Pickut • Patrick Cras

Parkinson’s disease (PD) is the second most common neurodegenerative brain disorder and is characterized by motor symptoms such as tremor, bradykinesia, rigidity and postural instability. A majority of the patients also develop non-motor symptoms. Impulse control disorders (ICD) are behavioural changes that often fail to be detected in clinical practice. The prevalence of ICD in PD varies widely from 6.1 to 31.2 % and treatment with dopaminergic medication is considered to be the greatest risk factor. Management consists mainly of reducing dopaminergic medication. In our experience, ICD has a tremendous impact on the quality of life of the patients and their families and should therefore not be disregarded. Studies addressing the role of ICD in PD caregiver strain are imperative. We attempt to give a comprehensive overview of the literature on the complicated neurobiology of ICD and discuss risk factors, genetic susceptibility, screening modalities and management.

ICD that has been described as a complication of PD includes pathological gambling (PG), hypersexuality, compulsive eating and buying [5]

Hypersexuality is observed in 1.92–8.9 % of PD patients on dopamine receptor agonist therapy vs. 1 % in the general population [6, 10–12, 14–18]. The behaviour may present itself as a preoccupation with sexual thoughts, demands for sex, desire for frequent genital stimulation, promiscuity, habitual use of sex telephone lines and internet pornography or contact with sex workers. The occurrence of paraphilias and change in sexual preference have also been described [23–25].

Studies about naltrexone, another opioid antagonist, and N-acetylcysteine, a glutamate-modulating agent, suggest they are both efficacious in the treatment of PG in general population [109, 111]. However, it is advised to interpret the results with caution due to the small number of participants.

Seedat S et al (2000) Pathological gambling behaviour: emergence

secondary to treatment of Parkinson’s disease with

dopaminergic agents. Depress Anxiety 11(4):185–186

 Bostwick JM et al (2009) Frequency of new-onset pathologic

compulsive gambling or hypersexuality after drug treatment of

idiopathic Parkinson disease. Mayo Clin Proc 84(4):310–316

Frequency of New-Onset Pathologic Compulsive Gambling or Hypersexuality After Drug Treatment of Idiopathic Parkinson Disease

  1. MICHAEL BOSTWICK, MD; KATHLEEN A. HECKSEL, MD; SUSANNA R. STEVENS, MS; JAMES H. BOWER, MD; AND J. ERIC AHLSKOG, MD, PHD

OBJECTIVE: To determine the frequency of new-onset compulsive gambling or hypersexuality among regional patients with Parkinson disease (PD), ascertaining the relationship of these behaviors to PD drug use. PATIENTS AND METHODS: We retrospectively reviewed the medical records of patients from 7 rural southeastern Minnesota counties who had at least 1 neurology appointment for PD between July 1, 2004, and June 30, 2006. The main outcome measure was compulsive gambling or hypersexuality developing after parkinsonism onset, including the temporal relationship to PD drug use. RESULTS: Of 267 patients with PD who met the study inclusion criteria, new-onset gambling or hypersexuality was documented in 7 (2.6%). All were among the 66 patients (10.6%) taking a dopamine agonist. Moreover, all 7 (18.4%) were among 38 patients taking therapeutic doses (defined as ≥2 mg of pramipexole or 6 mg of ropinirole daily). Behaviors were clearly pathologic and disabling in 5: 7.6% of all patients taking an agonist and 13.2% of those taking therapeutic doses. Of the 5 patients, 2 had extensive treatment for what was considered a primary psychiatric problem before the agonist connection was recognized. CONCLUSION: Among the study patients with PD, new-onset compulsive gambling or hypersexuality was documented in 7 (18.4%) of 38 patients taking therapeutic doses of dopamine agonists but was not found among untreated patients, those taking subtherapeutic agonist doses, or those taking carbidopa/levodopa alone. Behaviors abated with discontinuation of agonist therapy or dose reduction. Because this is a retrospective study, cases may have been missed, and hence this study may reflect an underestimation of the true frequency. Physicians who care for patients taking these drugs should recognize the drug’s potential to induce pathologic syndromes that sometimes masquerade as primary psychiatric disease. Mayo Clin Proc. 2009;84(4):310-316

Therefor the manuscript is not well prepared. Generally points:

Generally, the ms should contain a List of Abbreviations.

  1. Introduction

The Introduction need a substantial improvement:

Please add references at the end of this sentence: Impulse control disorders (ICDs) are frequently found in patients treated for Parkinson’s disease (PD) and are now known to be a relatively frequent side effect of dopamine replacement therapy (DRT).

Please add references at the end of this sentence: ICDs include different impulsive behaviors, such as pathological gambling, hypersexuality, binge eating or compulsive shopping.

Please add references at the end of this sentence: They share common symptoms linked to the field of addiction, particularly the failure to reduce or control a behavior despite impaired functioning and the resulting negative impact on the quality of life.

Please add references at the end of this sentence: The lifetime prevalence of hypersexuality was found to be 2.7% in PD patients on DRT and 7.4% in PD patients treated with dopamine agonists (DAAs) [1] and is probably underestimated because of the difficulty of addressing this topic for many patients.

Please add after [2] and [3] more references: There is no current approved pharmacological treatment for ICDs in PD patients, but a few studies have focused on the effectiveness of opioid antagonists approved for alcohol dependence treatment in the treatment of ICDs in PD patients [2], especially those with gambling disorders [3].

Please describe more exactly all studies with corresponding results which your referred in [2] and [3]. See the sentence: There is no current approved pharmacological treatment for ICDs in PD patients, but a few studies have focused on the effectiveness of opioid antagonists approved for alcohol dependence treatment in the treatment of ICDs in PD patients [2], especially those with gambling disorders [3].

Please add after [4] more references: The effectiveness of these drugs seems to be linked to a genetic polymorphism of the mu receptor, A118G, although that is still debated [4].

Please say: Mu opioid (Mu) receptor. See the sentence: The effectiveness of these drugs seems to be linked to a genetic polymorphism of the mu receptor, A118G, although that is still debated [4].  

Please add references at the end of this sentence: Indeed, the mesocorticolimbic dopamine pathway, which is known to mediate the reward system, has been implicated in the neurobiology of addiction, and the involvement of DRT in the emergence of ICDs in PD patients reinforces this hypothesis.

Please add references at the end of this sentence: The opioid system can modulate dopaminergic pathways. Current evidence points to overlaps between addictions, i.e., neurobiological mechanisms, epidemiology, comorbidities, genetic contributions, etc., and leads to the concept of pan-addiction treatment.

  1. Materials and Methods

Please change all headlines number like this:

2.1. Search strategy

2.2. Eligibility criteria

2.3. Article selection

2.4. Data extraction

Search strategy:

Please add more exactly instead of “from inception” the date or  month and year when you started your searching. See the sentence: A systematic review of the available literature was conducted to identify all relevant publications using PubMed and ScienceDirect from inception to January 2020.

Figure 1. Search strategy:

For better understanding and better legibility, please add below the Figure 1. as a Legend the briefly description of your search strategy.

  1. Results
  2. Systematic review

Please add to Systematik review section: how many patients were female, male and the mean age of all patients. 

Please explain, why you said: “Of the potential 556 articles” and you showed in your Figure 1: Pubmed (n=202) and Sciences Direct (n=395)? Together the two are 597 publications and not 556 publications?

Is it correct? Please explain or correct in manuscript.   This is very confusing for readers.

Table 1:

Has nothing to do with PD + medication – perhaps add as supplement

  1. Case presentation

Please say instead of “Mr. A is a 66-year-old Caucasian male”: Patient A. is a 66-year-old Caucasian male. Because both Mr and male is repeated unnecessarily.   Please insert abbreviation for Levodopa (L-Dopa) and use further in the whole manuscript. See sentence: Treatment with rasagiline, a selective monoamine oxidase (MAO) B inhibitor, was added in 2010 to 10 enhance the dopaminergic correction. This treatment was stopped in 2012, and additional L-11 dopa/benserazide therapy was initiated instead, as seen in our treatment chart (Figure 1).    4. Discussion Please add references at the end of this sentence: Indeed, there are several previous reports of this adverse drug reaction in the literature, and a statistically significant association was found between DAAs and ICDs, including hypersexuality. Please add references at the end of this sentence: Actually, there is no recommended pharmaceutical treatment for ICDs or hypersexuality. Please add more references at the end of this sentence: In the case of hypersexuality induced by DRT, the first step is to reduce the dose of dopaminergic medication and, if possible, to switch from DAAs to levodopa [9]. Please add references at the end of this sentence: The concept of treating ICDs with opioid antagonists is based on the neurobiological similarities between addictive disorders in general and substance use disorders. Please add references at the end of this sentence: Most often, patients respond to dopaminergic therapy reduction, but sometimes the DRT cannot be modified, or the symptoms persist despite the adjustment. Please add references at the end of this sentence: The role of the mesocorticolimbic dopaminergic pathway in the reward system has been widely described and strongly implicated in addiction. Please add references at the end of this sentence: The opioid system can modulate dopaminergic pathways: we know of three opioid receptors, the mu opioid receptors (MORs), the delta opioid receptors (DORs) and the kappa opioid receptors (KORs), that are activated by endogenous opioid peptides and by recreational drugs. Please add references at the end of this sentence: The use of an opioid antagonist for the treatment of ICDs in the course of treatment for PD should be initiated with care because of its impact on dopamine levels. Please add references at the end of this sentence: Indeed, PD is caused by a dopamine 127 deficit in the substantia nigra and the striatum. Please add references at the end of this sentence: The purpose of PD treatments such as DAAs or levodopa is to increase dopamine levels in the nigrostriatal pathway, whereas the effect of opioid antagonists is to ultimately decrease dopamine levels. Please add into Discussion and discuss after these both sentence: Indeed, PD is caused by a dopamine deficit in the substantia nigra and the striatum. The purpose of PD treatments such as DAAs or levodopa is to increase dopamine levels in the nigrostriatal pathway, whereas the effect of opioid antagonists is to ultimately decrease dopamine levels:  Please add references at the end of this sentence: Interindividual variations can be influenced by several factors, including genetic background. Please add references at the end of this sentence: OPRM1 has a 139 functionally significant and common variant termed A118G (rs1799971). Please add references at the end of this sentence: In turn, A118G causes amino acid exchange at residue 40 of the MOR from the normal asparagine 142 (Asn, A allele) to an abnormal aspartic acid (Asp) residue (G allele) (Asn40Asp). Please add references at the end of this sentence: A significant reduction in 145 the effectivity of signaling pathways after the binding of specific agonists has been observed. 

Please add to discussion: What about the serious side-effects of anti-craving drugs such as severe liver and kidney dysfunction?  What does that mean for patients with hypersexuality treated with this medication?

Round 2

Reviewer 2 Report

The revision was done intensively. All but one comments of the reviewer were fullfilled. The ms is massively improved.

Unfortunately, the authors insist to include their Tab 1 in the MS and not as a supplement.